# The Pathophysiological Significance of Fibulin-3

**DOI:** 10.3390/biom10091294

**Published:** 2020-09-08

**Authors:** Imogen Livingstone, Vladimir N. Uversky, Dominic Furniss, Akira Wiberg

**Affiliations:** 1Nuffield Department of Orthopaedics, Rheumatology and Musculoskeletal Sciences, University of Oxford, Botnar Research Centre, Nuffield Orthopaedic Centre, Oxford OX3 7LD, UK; imogen.livingstone@jesus.ox.ac.uk (I.L.); dominic.furniss@ndorms.ox.ac.uk (D.F.); 2Laboratory of New Methods in Biology, Institute for Biological Instrumentation, Russian Academy of Sciences, Federal Research Center “Pushchino Scientific Center for Biological Research of the Russian Academy of Sciences”, Pushchino 142290, Moscow Region, Russia; vuversky@usf.edu; 3Department of Molecular Medicine, Morsani College of Medicine, University of South Florida, Tampa, FL 33612, USA; 4Department of Plastic and Reconstructive Surgery, Oxford University Hospitals NHS Foundation Trust, John Radcliffe Hospital, Oxford OX3 9DU, UK

**Keywords:** fibulin-3, EFEMP1, extracellular matrix, connective tissue, genome-wide association study, intrinsically disordered protein, protein–protein interactions

## Abstract

Fibulin-3 (also known as EGF-containing fibulin extracellular matrix protein 1 (EFEMP1)) is a secreted extracellular matrix glycoprotein, encoded by the *EFEMP1* gene that belongs to the eight-membered fibulin protein family. It has emerged as a functionally unique member of this family, with a diverse array of pathophysiological associations predominantly centered on its role as a modulator of extracellular matrix (ECM) biology. Fibulin-3 is widely expressed in the human body, especially in elastic-fibre-rich tissues and ocular structures, and interacts with enzymatic ECM regulators, including tissue inhibitor of metalloproteinase-3 (TIMP-3). A point mutation in *EFEMP1* causes an inherited early-onset form of macular degeneration called Malattia Leventinese/Doyne honeycomb retinal dystrophy (ML/DHRD). *EFEMP1* genetic variants have also been associated in genome-wide association studies with numerous complex inherited phenotypes, both physiological (namely, developmental anthropometric traits) and pathological (many of which involve abnormalities of connective tissue function). Furthermore, *EFEMP1* expression changes are implicated in the progression of numerous types of cancer, an area in which fibulin-3 has putative significance as a therapeutic target. Here we discuss the potential mechanistic roles of fibulin-3 in these pathologies and highlight how it may contribute to the development, structural integrity, and emergent functionality of the ECM and connective tissues across a range of anatomical locations. Its myriad of aetiological roles positions fibulin-3 as a molecule of interest across numerous research fields and may inform our future understanding and therapeutic approach to many human diseases in clinical settings.

## 1. Introduction

Fibulin-3, also known as EGF-containing fibulin extracellular matrix protein 1 (EFEMP1), is an extracellular matrix glycoprotein encoded by the *EFEMP1* gene that has emerged over the past two decades as a molecule of increasing significance in human health and disease. Although its definitive function remains incompletely understood, a growing body of evidence implicates fibulin-3 in a myriad of pathophysiological processes, ranging from ophthalmic disease to cancer metastasis to complex inherited traits. The far-reaching pathogenesis of this molecule is both exciting and challenging, with the functional roles of fibulin-3 being elucidated from a backdrop of ever-increasing complexity. As such, this review aims to collate our existing understandings regarding the physiological and pathological processes associated with fibulin-3, and thereby to synthesise a clearer picture of its potential mechanistic roles in various complex pathologies. In doing so, we aim to highlight the potential significance of this molecule both in our understanding of disease and in our subsequent approach to therapeutics in clinical settings.

## 2. Fibulin-3 Overview

### 2.1. Structural Characterisation

Fibulin-3 was first described in 1995 as a protein containing epidermal growth factor (EGF)-like domains encoded by an mRNA transcript termed S1-5 that is overexpressed in human Werner syndrome fibroblasts, an inherited condition of early aging characterised by premature cellular senescence [1,2]. Further analysis of the S1-5 transcript at the genomic and protein levels led to its classification as a novel member of the fibulin family (fibulin-3) [3,4], based on its sequence homology and predicted structural homology to fibulins 1 and 2, including the presence of some characteristic fibulin features, such as a fibulin-type C-terminal globular domain and a tandem array of central calcium-binding EGF-like modules [3]. However, its shorter length and unique N-terminal interrupted EGF-like module earned fibulin-3 a sub-categorisation as a ‘short fibulin’ [3] (a group now containing fibulins 3, 4, 5, and 7), as distinct from the ‘long fibulins’ (fibulins 1, 2, 6, and 8) [5] (Figure 1).

Despite substantial structural overlap with other short fibulins, accumulating evidence suggests that fibulin-3 may fulfil distinct biological functions. Definitive biochemical characterisation of the short fibulins, including murine fibulin-3, confirmed its structural homology to other fibulins, including the short rod structure with a globule at one end typical of a protein with tandem EGF-like modules [6]. Alternative splicing of *EFEMP1* may give rise to fibulin-3 isoforms. Five human *EFEMP1* splice variants differing in their N-terminal sequence have been identified, of which the largest and smallest are significantly expressed at the protein level [1]. Immunoblotting studies of murine fibulin-3 have also identified two differently sized protein bands, suggesting different fibulin-3 isoforms could indeed exist physiologically [6]. Murine fibulin-3 also has the capacity for extensive post-translational modification including N- and O-glycosylation [6], something that is not well studied in the human protein, but which represents another possible basis for fibulin-3 isoforms. How these pre- and post-translational modifications modulate the function of fibulin-3 remains relatively unexplored. However, given the multitude of *EFEMP1* genetic variants that have been associated with a range of human pathophysiological traits (discussed below), it seems feasible that such associations could be underpinned by the functional variation of splice variants or distinctly glycosylated forms of fibulin-3.

It is likely that the multifunctionality and polypathogenicity of human fibulin-3 might be related to its ability to be engaged in multiple interactions with various partners. In agreement with this hypothesis, the IntAct database (http://www.ebi.ac.uk/intact/) of binary interactions [7] indicated that human fibulin-3 is engaged in 157 binary interactions. Binding promiscuity of this protein is further illustrated by Figure 2, which represents the results of this protein analysis by the STRING computational platform (Search Tool for the Retrieval of Interacting Genes; http://string-db.org/) generating protein–protein interaction (PPI) networks based on the predicted and experimentally-derived information on the interaction partners of a protein of interest [8]. In fact, Figure 2 shows that the human fibulin-3-centered network contains 58 nodes (proteins) connected by 283 edges (PPIs). In this network, the average node degree is 10.1 (i.e., on average each member of this PPI has 10.1 interactions with other PPI members), and its average local clustering coefficient is 0.75. The local clustering coefficient defines how close its neighbors are to being a complete clique; it is equal to 1 if every neighbor connected to a given node Ni is also connected to every other node within the neighborhood, and it is equal to 0 if no node that is connected to a given node Ni connects to any other node that is connected to Ni. Since the expected number of interactions among proteins in a similar-sized set of proteins randomly selected from human proteome is equal to 82, this PPI network has significantly more interactions than expected, being characterised by a PPI enrichment *p*-value of <10^−16^. Therefore, this analysis clearly defines human fibulin-3 as a hub protein.

All these observations (binding promiscuity and the presence of alternatively spliced isoforms and multiple posttranslational modification (PTM) sites) provide an important hint towards the potential intrinsically disordered nature of human fibulin-3. In fact, intrinsically disordered proteins (IDPs) or intrinsically disordered protein regions (IDPRs) are functional proteins or protein regions that lack ordered 3D structures [9,10,11,12,13,14,15,16,17,18,19]. These proteins have the ability to bind to multiple partners, which enables them to function in regulation, signalling, and control, where they are commonly engaged in one-to-many and many-to-one interactions [9,11,15,16,17,20,21,22,23,24,25,26]. Disordered proteins or protein regions are often affected by PTMs, such as phosphorylation, glycosylation, methylation, and ubiquitylation [27,28], and serve as major targets for the alternative splicing [29,30,31]. All these processes are utilised by nature to control and regulate functions of IDPs or hybrid proteins containing ordered domains and functional IDPRs. However, deregulation of IDPs is dangerous and these structure-less, highly dynamic, promiscuously interacting proteins/regions are implicated in numerous human diseases [32,33,34]. Since multifunctionality and binding promiscuity are typically rooted within the protein intrinsic disorder phenomenon, we looked here at the intrinsic disorder propensities of human fibulin-3 and at the roles of intrinsic disorder in some of its physiological functions.

Figure 3 supports this view by showing some of the disorder-related features of human fibulin-3. In fact, Figure 3A illustrates that this protein contains several intrinsically disordered and flexible regions (i.e., regions with predicted disorder scores (PDS) ≥ 0.5 and 0.15 ≤ PDS < 0.5, respectively). Since fibulin-3 contains a large number of cysteine residues engaged in the formation of 15 disulfide bridges concentrated within its tandem array of calcium-binding EGF-like modules, we also looked for the presence of redox-sensitive regions (i.e., cysteine-containing regions capable of the disorder-to-order or order-to-disorder transitions associated with changes in the redox state of the environment [35]) in this protein. Figure 3B clearly indicates that human fibulin-3 contains two such redox-sensitive IDPRs, residues 19–66 and 139–385. Finally, to gain more information on the potential functional roles of IDPRs in human fibulin-3, this protein was subjected to the disorder analysis using the D^2^P^2^ platform (http://d2p2.pro/) [36], which is a database of predicted disorders for proteins from completely sequenced genomes [36]. D^2^P^2^ uses outputs of several per-residue disorder predictors, such as IUPred [37], PONDR^®^ VLXT [38], PrDOS [39], PONDR^®^ VSL2 [40,41], PV2 [36], and ESpritz [42]. The database is further supplemented by the data on the locations of predicted SCOP (Structural Classification of Proteins) domains, conserved Pfam domains, as well as sites of various posttranslational modifications and predicted disorder-based protein binding sites, known as molecular recognition features (MoRFs) [36].

It is known that many disorder-based binding regions are characterised by the presence of less disordered sub-regions, which are not capable of folding on their own, but can undergo binding-induced folding upon interaction with a binding protein partner. In disorder profiles, such regions are typically manifested as local “dips” within regions with high disorder scores [46,47]. In D^2^P^2^, the presence of MoRFs is evaluated by the ANCHOR algorithm [48,49]. Figure 3C shows that human fibulin-3 is predicted to have one MoRF and several phosphorylation sites, mostly located within disordered or flexible regions. Therefore, these data indicate that conformational flexibility and intrinsic disorder of human fibulin-3 are important functional features of this protein and might serve as a foundation for its multifunctionality and polypathogenicity.

### 2.2. Fibulin-3 Expression Patterns

Analysis of fibulin-3 expression patterns in both human tissue (see Figure 4) and mouse tissue has yielded some valuable insights into its potential physiological functions. Such functions are likely to exhibit at least some degree of conservation between these two species, given the high level of sequence identity between the human and murine fibulin-3 homologues [50].

#### 2.2.1. Adult Tissue

In adult human tissue, fibulin-3 mRNA is widely expressed, having been detected in the heart, lung, skeletal muscle, stomach, pancreas, liver, small and large intestine, spleen, kidney, prostate gland, testis, ovary, placenta, and brain (at low levels) [3,51]. Fibulin-3 protein is abundantly expressed in many human ocular tissues, particularly in the ciliary body and choroid retinal pigment epithelium (RPE) [52,53,54,55]. It has also been detected in human foetal membranes, specifically in amnion epithelial cells and chorion trophoblast cells [56].

In adult mouse tissue, fibulin-3 mRNA is most abundant in the lung; readily detectable in ovary, kidney, and skeletal muscle; present at low levels in the liver and stomach; and undetectable in the spleen, heart, brain, and intestine [1]. It is also strongly expressed in the eye [50]. Fibulin-3 protein is preferentially located in the walls of the vasculature, in particular in capillaries [3], medium-sized vessels [3], and the aorta [6], but it is absent from large myocardial vessels and heart valves [3]. Murine fibulin-3 is also expressed in the oesophagus and lung [6] (especially around bronchiole basement membranes [3]). Notably, these are all tissues enriched with elastic fibres. It is also moderately expressed in the stomach, intestine, skeletal muscle, and testis; and is present at low levels in the spleen, placenta, skin, thymus, heart, and kidney (in both renal blood vessels and peritubular regions [3]) [6].

#### 2.2.2. Embryonic Tissue

Fibulin-3 mRNA expression patterns in embryonic mouse tissue strongly implicate the protein in bone and cartilage development. During embryonic development, murine fibulin-3 mRNA expression initiates at E9.5 in condensing mesenchyme structures (somites and the paraxial mesenchyme of the cranial and proximal trunk regions), and persists throughout embryogenesis in the derivative bone and cartilaginous structures from these areas, including the craniofacial skeleton arising from the 2nd and 3rd branchial arches, the appendicular skeleton derived from limb bud cartilage rudiments, and the axial skeleton formed by developing vertebrae [50]. Fibulin-3 protein expression has subsequently been confirmed in the perichondrium of E15 embryonic mouse sections [6], substantiating a role for this protein in skeletal development. Fibulin-3 protein is also expressed in developing lungs, both in the walls of the lung vasculature and in the basement membranes of the large airways [6], suggesting that fibulin-3 has a role in both the developing and definitive adult lung [6]. In contrast, despite being abundant in adult human ocular tissue [52,53,54,55], fibulin-3 expression is absent from embryonic ocular structures [50], so it does not seem to be implicated in ocular development.

### 2.3. Protein-Protein Interactions

Figure 2 already showed that fibulin-3 is a promiscuous binder. The protein interaction profile of fibulin-3 is strikingly distinct from that of other members of the fibulin family, substantiating the notion that fibulin-3 fulfils a physiological function distinct from that of other members of the family. Short fibulins have a more limited protein binding profile than that of fibulins 1 and 2, preferentially interacting with elastic fibre-related proteins, as opposed to additional basement membrane components and fibronectin, and fibulin-3 seems particularly restricted in this regard [6].

#### 2.3.1. Tropoelastin

All fibulins are localised in tissues rich in elastic fibres, consistent with their interaction with tropoelastin (the monomer of elastic fibrils) and substantiating a role in elastic fibre homeostasis. Unlike other fibulins, fibulin-3 only weakly interacts with tropoelastin [3,6], to which fibulins 2 and 5 bind strongly, and fibulins 1 and 4 moderately [6]. However, their criticality to elastogenesis in vivo, and their relative abundance in these elastic tissues, does not directly correlate with binding strength in vitro, suggesting that there may be a degree of functional compensation between some of the family members. Fibulins 4 and 5 are essential for elastic fibre formation. *Fbln4* knock out (KO) in mice abolishes elastogenesis and is embryonically lethal [57], and *Fbln5* KO mice exhibit systemically disorganised elastic fibre networks and phenotypic features of defective elastic tissue, including altered vascular compliance, emphysema, and cutis laxa [58]. Lack of fibulins 1 or 2 does not perturb elastic fibre integrity [6]. Fibulin-3 deficiency has a much more specific effect, leading to an elastic fibre deficit localised to fascia connective tissue [59]. This phenotypically manifests as defects associated with fascia weakness (such as herniation), whereas the functionality of other elastic fibre-containing organs where fibulin-3 is known to be highly expressed, such as the lungs and vasculature, remains intact, with only occasional reports of loose skin [59]. Moreover, there is no evident redundancy for fibulin-3 functionality in these animals, with the presence of neither fibulin-4 nor -5 compensating for lack of fibulin-3, further distinguishing fibulin-3 as a functionally unique member of the fibulin family that holds a distinct, more confined, role in elastic fibre biology.

Indeed, the distinct tropoelastin interaction profile of fibulin-3 is consistent with, and may even underpin, its different expression patterns and functional manifestations compared to the other short fibulins, in particular, the relative insignificance of fibulin-3 to the structural integrity of elastic blood vessels. Whilst the other fibulins form a protective layer around elastin within these vessel walls, helping to maintain their structure [6,57,58], fibulin-3 only weakly interacts with tropoelastin [3,6] and is less abundantly expressed in large elastic vessels. In mice, fibulin-5 is substantially more abundant in the aorta [6], and fibulin-4 is prominent in the heart valves, aortic media, and large myocardial vessels from which fibulin-3 is absent, instead being present mainly in myocardial capillaries [3]. Moreover, embryonic expression distributions also point towards a lesser role for fibulin-3 in elastic blood vessels. Fibulin-5 mRNA expression initiates concurrently with fibulin-3 at E9.5 [50,60], but is mainly present in rudimentary structures of the cardiovascular system, as opposed to the musculoskeletal system, becoming prominent in cardiac primordia, including the endocardial cushions, cardiac outflow tract, and the aorta from E12.5, where it remains strongly expressed throughout embryogenesis [60]. These distinct developmental roles are consistent with their streamlined definitive roles in the adult—from the outset, fibulin-3 functionality appears to be more confined to elastic connective tissue structures outside the cardiovascular system.

#### 2.3.2. Matrix Metalloproteinases (MMPs) and Tissue Inhibitors of Metalloproteinases (TIMPs)

It may be that this regionalised influence of fibulin-3 on elastic fibre integrity arises indirectly, involving a fibulin-3 interaction with another molecule that secondarily influences elastic fibres in a tissue-specific manner [59]. A candidate molecule fulfilling such a link is TIMP-3, a member of the family of endogenous tissue inhibitors of extracellular matrix (ECM) matrix metalloproteinases (MMPs), with which fibulin-3 strongly interacts [61]. Together the TIMP and MMP enzyme groups maintain the integrity of ECM components by preserving the equilibrium between ECM degradation and synthesis [62]. It has been postulated that fibulin-3 could protect elastic fibres from degradation by antagonising the responsible ECM proteases, either by enhancing TIMP-3-mediated MMP inhibition or through directly attenuating the activity of specific elastin-degrading MMPs, and that the specific confinement of this protective role to fascia could arise from the regionalisation of the molecules with which it interacts [59].

Fibulin-3 influences the expression of certain TIMPs and MMPs in vitro. Murine microvascular endothelial cells transfected with fibulin-3 exhibit reduced MMP2 and MMP3 mRNA levels, and increased levels of TIMP-1 and TIMP-3 mRNA, manifesting as reduced MMP2 activity, implicating fibulin-3 as a suppressor of ECM proteolysis/degradation [63]. Furthermore, fibulin-3 knock-down in mouse embryonic fibroblasts is associated with increased cell invasiveness, which could be attributed to the elevation of MMP9 activity also seen in these cells, suggesting fibulin-3 may have a role in suppressing the ECM turnover that is required for cell invasion via MMP inhibition [64]. However, it remains uncertain whether such a role is preserved in vivo, occurs in other cell types such as human fascial fibroblasts, and also occurs at the protein level. There are some data from human epithelial tissue to support the existence of a reciprocal relationship between fibulin-3 and MMPs that could regulate tissue integrity. TNF induces a dose-dependent weakening of foetal membrane fragments in vitro, which was associated with a reduction in amnion cell fibulin-1, -3 and -5 levels, and a simultaneous increase in MMP levels (MMP9 in amnion epithelial cells and MMP2 in amnion mesenchymal cells) [56]. Importantly, these in vitro data were corroborated by the finding that fibulin-3 protein levels were less abundant in the ‘weak zone’ of the human foetal membranes obtained at the point of delivery [56], highlighting another region of the human body in addition to the fascia where a lack of fibulin-3 appears to influence tissue strength, implicating it as an important functional contributor to the integrity of specific anatomical structures.

#### 2.3.3. ECM1

Fibulin-3 interacts with extracellular matrix protein 1 (ECM1), a secreted glycoprotein, with which it colocalises in human skin [65]. Both fibulin-3 and ECM1 have roles in angiogenesis, endochondral bone formation, and tumour biology, suggesting a degree of functional overlap between these two glycoproteins [65]. However, the biological significance of this interaction has not been investigated further.

## 3. Mouse Model of Fibulin-3 Deletion

Loss of fibulin-3 in mice via targeted disruption of the murine *Efemp1* gene leads to the development of a complex phenotype that provides useful insights into the biological functions of fibulin-3 [59]. The primary defect in *Efemp1* KO mice is a deficiency of elastic fibres specifically within fascia tissue, causing a phenotype of multiple herniations, including lateral bulges reflecting direct inguinal hernias, anorectal bulges reflecting indirect inguinal hernias in males and pelvic organ prolapse in females, and thoracic bulges representing protrusion of the xiphisternum through the fascial tip of the linea alba. The functionality of other elastic fibre-rich organs with high fibulin-3 expression levels remains intact, with only occasional reports of loose skin in these animals, but no evident lung or vascular defects. Notably, the herniation phenotype is influenced by the genetic background of the mice, with those on a C57BL/6 background developing numerous large hernias from two months of age onwards, whereas the BALB/c background mice rarely exhibited herniation, and the few cases that did occur were exclusively anorectal herniae in mice of more advanced age [59]. The authors postulated whether this reflected the presence of fibulin-3 functional modulators in the different mouse strains in the form of genes that either compensate for fibulin-3 functionality (in BALB/c mice) or that promote herniation and are normally suppressed by fibulin-3 (in C57BL/6 mice). As discussed, the highly distinct phenotype arising from the loss of *Efemp1* compared to that of *Fbln4* or *Fbln5* substantiates the case that fibulin-3 is a functionally unique member of the fibulin family with a more localised role in elastic fibre biology.

Other features of these fibulin-3-deficient mice, irrespective of genetic background, included reduced reproductivity; reduced lifespan (median 93 vs. 115 weeks); and some characteristics of early-onset aging, including lower body mass, lordokyphosis, reduced hair growth, and a systemic atrophy of adipose tissue, muscle, and internal organ tissue [59].

## 4. Fibulin-3 In Human Disease

### 4.1. Disorder of Mutant Fibulin-3: Malattia Leventinese/Doyne Honeycomb Retinal Dystrophy

A specific non-conservative mutation in the *EFEMP1* gene causes an inherited autosomal dominant macular degeneration called Malattia Leventinese (ML) (otherwise known as Doyne honeycomb retinal dystrophy (DHRD)), characterised by the formation of drusen (pathological deposits of ECM) between the RPE and Bruch’s membrane in the eye [66]. This is of clinical interest given the apparent phenotypic overlap between ML/DHRD and the common sporadic condition of age-related macular degeneration (AMD).

Various lines of evidence suggest that the mutation induces disease through a gain-of-function mechanism. Firstly, ML/DHRD exhibits autosomal dominant inheritance [66]. Secondly, *Efemp1* KO mice do not develop an early-onset macular degeneration phenotype [59], making it unlikely that loss of fibulin-3 triggers the pathology. Thirdly, mutant fibulin-3 misfolds and is poorly secreted by RPE cells, leading to accumulation within RPE cells and between the RPE and drusen deposits in the eyes of ML patients—a site at which fibulin-3 is normally absent in healthy eyes [67]—thus causally implicating the protein based on its aberrant localisation with the site of disease.

Interestingly, fibulin-3 expression is not detected in the developing eye in mice embryos [50], but is highly expressed in the adult eye in both mice [50,66] and humans, in particular in cells of the RPE [53,54,55,66]. It therefore remains unclear at which stage of life fibulin-3 expression assumes an important role in the RPE. Ultimately, there remains a lack of insight into how the fibulin-3 mutation in ML/DHRD causes the initial aberrant ECM deposition and drusen formation that are central to early-onset macular degeneration. These are not simply accumulations of mutant fibulin-3, which is in fact only a minor component of drusen itself [67]. The key question remains as to whether they arise secondarily to RPE dysfunction and perturbation of its normal secretory profile; or as a primary signalling effect of the mutant fibulin-3, exerted either intra- or extracellularly in an autocrine or paracrine manner.

One mechanistic pathway that has been proposed involves induction of intracellular endoplasmic reticulum (ER) stress and activation of the unfolded protein response (UPR) by the accumulation of misfolded fibulin-3 in the ER of RPE cells [68]. The UPR is an adaptive multi-armed cellular signalling cascade that facilitates ER protein clearance and degradation pathways and hence enhances the capacity of the ER to cope with its increased protein load [69]. The UPR is a process that could feasibly drive recognised features of ML pathology, including both RPE dysfunction, apoptosis and eventual atrophy, as well as the upregulation of pro-angiogenic VEGF, which may underpin choroidal neovascularisation, which contributes to the impairment of vision [68].

The UPR is potently induced by mutant fibulin-3 in ARPE-19 cells [68]; indeed, certain structural components of fibulin-3 make it a feasible candidate for induction of ER stress. Its numerous calcium-biding sites within the EGF-like modules could, in the context of accumulated misfolded fibulin-3 within the ER, lead to substantial calcium sequestration and depletion of ER free calcium, a potent UPR inducer [70]. Moreover, its many intramolecular disulphide bonds could form abnormally and propagate a detrimental cycle of bond reduction, glutathione consumption, and reactive oxygen species production, which also drives ER stress and apoptosis [71]. Substantiating a role for mutant fibulin-3-induced ER stress in the pathogenesis of ML is more recent evidence that fibulin-3 has numerous intracellular interacting partners, the majority of which are involved in protein folding or protein degradation in the ER [72], thus confirming the feasibility that part of the pathological impact of mutant fibulin-3 could be the perturbation of normal cellular proteostasis.

### 4.2. Genome-Wide Association Studies (GWAS) Associations

Fibulin-3 is also implicated by association in numerous complex phenotypes with polygenic heritable components. Genome-wide association studies (GWASs) have found *EFEMP1* genetic variants to be significantly associated with phenotypes that can broadly be categorised into anthropometric (adult height, pre-pubertal growth, forced vital capacity); ocular (glaucoma endophenotypes—intraocular pressure (IOP), optic cup area (OCA), vertical cup disc ratio (VCDR)); cardiovascular (chronic venous disease, blood pressure); neurological (carpal tunnel syndrome); gastrointestinal (biliary atresia, diverticular disease); musculoskeletal (inguinal hernia, joint hypermobility); immunological (childhood ear infection); and neoplastic (glioma, breast cancer) (Figure 5). Fibulin-3 expression has been detected, and in some cases is upregulated, in the relevant disease tissue of interest across many of these phenotypes, further supporting a putative aetiological role of fibulin-3 in these traits based on spatial proximity. Importantly, there is evidence to suggest that many of these phenotypes could be underpinned by an abnormality in connective tissue function.

#### 4.2.1. Anthropometric Traits

The association between *EFEMP1* and numerous anthropometric traits substantiates its developmental role. *EFEMP1* genetic variants have consistently been associated with adult height across multiple populations, including individuals of European [73,74,75,76], African American [74], Scandinavian [73,74], Korean [77], Japanese [78], and Filipino [79] ancestry. Furthermore, some of the adult height *EFEMP1* variants (rs3791679, rs3791675) have also been found to associate with childhood height [80,81]. Indeed, a substantial portion of adult height variation attributable to a set (~180) of previously-reported single nucleotide polymorphisms (SNPs) [75] already seems to have manifested by age 10 (based on an allelic score system) [81], suggesting that the genetic component of height involves a combination of effects on pre-natal, pre-pubertal, and pubertal growth, some of which may involve fibulin-3. These human data thus strengthen the case that fibulin-3 has a role in skeletal development, first highlighted by fibulin-3 expression studies in murine embryos [6,50]. Interestingly, adult height reportedly has shared genetic predisposition with joint hypermobility, another trait with which *EFEMP1* is also associated [82], suggesting there may be some functional overlap in how *EFEMP1* influences each phenotype.

Other anthropometric parameters with which *EFEMP1* has been associated include waist circumference adjusted for BMI [83] and forced vital capacity (FVC) [84]. Gene set enrichment analysis of FVC-associated variants identified numerous pathways relevant to organ development and tissue/ECM remodelling, again pointing towards a plausible causal role of fibulin-3 in this trait. Fibulin-3 expression was confirmed in lung tissue, human bronchial epithelial cells, and human airway smooth muscle, and the specific *EFEMP1* variant (rs1430193) was identified as a likely effector of gene expression in human fibroblasts but not in lung tissue [84], supporting a role for fibulin-3 in FVC via connective tissue. Interestingly, the same *EFEMP1* variant has also been found in a phenome-wide association study (PheWAS) to be associated with standing height [85]. This adds to the notion that this protein has a broader role in determination of anthropometric traits, particularly given the pre-adjustment for height in the analysis of FVC-associated genes [84], meaning the observed FVC association is likely to be independent of height and instead more reflective of a more general genetic determination of organ size.

#### 4.2.2. Glaucoma Endophenotypes

*EFEMP1* variants are implicated by association in primary open angle glaucoma (POAG), an ophthalmic pathology with a relatively more complex aetiological genetic component compared to ML/DHRD [86,87]. Although GWASs have yet to associate *EFEMP1* genetic variants directly with POAG [52,88], they have been linked to various endophenotypes which themselves are associated with glaucoma and have a certain degree of heritability, including intraocular pressure (IOP) [52,89,90], the elevation of which is a well-established risk factor for POAG; optic cup area(OCA) [54,88]; and vertical cup-disc ratio (VCDR) [88].

Insight into the potential mechanistic underpinning of these GWAS *EFEMP1*–glaucoma associations may be gleaned from the studies’ concomitant gene and tissue enrichment analyses. Gene pathways enriched for loci associated with VCDR and OCA were related to cell differentiation, development, regulatory DNA binding, and Notch signalling [88], whereas those enriched with IOP loci included abnormal vascular endothelial cell morphology [89]. Importantly, these are pathways with which fibulin-3 has been linked in other contexts, including cell differentiation (as a negative regulator of chondrocyte differentiation [91]); developmental processes (namely murine bone and cartilage development [50]); Notch signalling in cancer, as an activator of pro-invasive [92] and pro-angiogenic [93] Notch signalling in glioma; and vascular endothelial cell activity in angiogenesis [63]. This suggests there may be a degree of mechanistic overlap between the role of fibulin-3 in these processes and its effect on the ocular phenotypes related to glaucoma.

Another IOP-associated pathway identified in a GWAS is extracellular matrix organisation [90]. ECM turnover in the trabecular meshwork (TM) has been posited as a regulator of IOP [94], and given that fibulin-3 is expressed in the TM [53,55], it is possible that fibulin-3 could exert a subtle influence on TM ECM turnover, which forms the biological basis for its link to IOP. Interestingly, ECM elasticity has been found to modulate the response of TM cells to TGF-β signalling in vitro, with increased elasticity promoting induction of protein expression patterns akin to those of POAG [95]. Thus, *EFEMP1* variants may produce changes in the elasticity of the ECM surrounding the TM, secondarily influencing that pathogenesis of POAG.

Another notable common finding across the literature is that loci associated with the *EFEMP1*-related glaucoma endophenotypes (IOP, VCDR, OCA) are relatively enriched in cell or tissue types that can be broadly classified as either musculoskeletal or connective tissue-related [88,89]. Additionally, IOP-related variants (including *EFEMP1*) located within known genomic enhancer regions have been found to correlate significantly with the promoter regions of nine genes in cell types that were all either stromal or ocular in anatomical location [52]. Given the clustering of likely functional IOP risk variants within stromal tissue, the authors postulated these genes could collectively contribute to some form of anterior iris stroma hypoplasia, which is established as the most common iris defect associated with developmental glaucoma [96], but could potentially also contribute to the pathogenesis of primary forms of the condition. Therefore, it is possible that the influence of fibulin-3 on these ocular phenotypes could be fundamentally explained by its role in connective tissue biology.

#### 4.2.3. Inguinal Hernia

The aetiology of inguinal hernia (IH) is known to have a metabolic component involving various components of the ECM, including collagen, elastin, and MMPs. It thus seems likely that the genetic risk of IH conferred by *EFEMP1* variants [97] is causally underpinned by some alteration of ECM biology within connective tissue. This is supported by *Efemp1* KO mice models, in which multiple hernias (including IH) develop secondary to deficiency of fascial elastic fibres [59], as well as the observation that fibulin-3 was the most highly expressed of the IH risk loci in mouse connective tissue [97]. Connective tissue dysfunction is also an aetiological factor in IHs that occurs in individuals with inherited syndromes of connective tissue disorder including cutis laxa, Marfan’s syndrome, and Ehlers–Danlos syndrome, all of which involve mutations in other ECM components [98]. It thus seems plausible that *EFEMP1* genetic variants could confer more subtle structural changes in the transversals fascia, which manifests as an increased propensity for development of IH.

Indeed, IH may be another context in which the increased risk of developing this phenotype is related to a shift in the relative balance in the activity of enzymes involved in ECM metabolism, namely MMPs and TIMPs. Gene regulatory network analysis identified *EFEMP1* interactions including elastin, collagen 15A1, and TIMP-3 [97]. Notably, TIMP-3 was a common protein of interest for *EFEMP1* and *WTI*, the other top IH risk loci with an evident link to connective tissue, encoding an ECM protein involved in connective tissue remodelling [97]. Given that TIMP-3 interacts with fibulin-3 [61] and is thought to be activated by WT1 [97], it may be that part of the increased risk of IH conferred by these variants lies in an alteration of TIMP-3 regulation that shifts the equilibrium between ECM degrading and synthesising enzymes and perturbs connective tissue homeostasis. In support of this idea is the observation that fibroblasts from IH patients have been found to have an imbalance between MMP and TIMP activity [99].

#### 4.2.4. Chronic Venous Disease

Fibulin-3 is expressed in venous vessel walls where it has been postulated that *EFEMP1* genetic variants functionally manifest as altered vein wall elasticity or reduced strength, which increases the propensity for the eventual development of chronic venous disease (CVD) and varicose veins (VVs) [100,101]. Interestingly, one of the specific *EFEMP1* variants (rs17278665) associated with CVD was predicted to cause a splice site alteration that could lead to loss of the region of the fibulin-3 protein that interacts with TIMP-3 [100], thus highlighting a putative pathway through which an imbalance in ECM enzymes could arise from specific *EFEMP1* genetic variants. Moreover, drug-target enrichment analysis indicated fibulin-3 is tractable to antibody targeting, positioning it as a possible therapeutic target in VVs [101].

It may also be that the aforementioned role of fibulin-3 in angiogenesis and vascular remodelling [63] is relevant in this context, with alteration of endothelial cell behaviour potentially contributing to the vessel wall pathology in CVD. It is unclear how such changes would be confined to the venous system; however, an *EFEMP1* variant (rs11899888) has been recently associated with both diastolic blood pressure and pulse pressure [102], suggesting that fibulin-3 may indeed have some degree of systemic influence over vascular compliance throughout the cardiovascular system.

#### 4.2.5. Diverticular Disease

Diverticular disease (DD) is characterised by the appearance of diverticula in the walls of the large intestine, in some ways analogous to a localised hernia. Genetic variants associated with DD were found to be enriched for expression in mesenchymal stem cells, many connective tissue cell types, and in mesenchymal development pathways [103]. *EFEMP1* was also a member of many of the gene pathways enriched for DD-associated loci that related to ECM biology, mesenchymal development, vascular morphology, and cell growth/proliferation. Moreover, the PheWAS results suggested DD had a common aetiology with herniation, providing some evidence for a direct role of altered connective tissue integrity that feasibly involves fibulin-3 in its pathogenesis [103]. A limitation in the interpretation of PheWAS findings is that these conditions could simply be a consequence of DD, as opposed to being driven by a common aetiology. However, given that *EFEMP1* has been strongly implicated in susceptibility to herniation independently of this study [59,97], a degree of overlap in aetiology converging on fibulin-3 in connective tissue is certainly a reasonable hypothesis.

#### 4.2.6. Carpal Tunnel Syndrome

Carpal tunnel syndrome (CTS) is a very common entrapment neuropathy of the hand caused by compression of the median nerve as it travels through a fibro-osseous tunnel in the wrist. A recent GWAS found a significant association between CTS and rs3791679, an intronic SNP that resides within an enhancer region of *EFEMP1.* Of the several genetic variants found to associate with CTS, many (including the *EFEMP1* locus) were relatively enriched in gene sets relating to ECM biology, and to anthropometric traits with which *EFEMP1* has previously been linked, namely, waist circumference [83] and height [76], suggesting there is a significant component of CTS pathology centered in ECM biology [104].

Interestingly, it seems the association of *EFEMP1* with both height and CTS may share an ECM-related mechanistic underpinning. Indeed, the most significant CTS-associated variant at the *EFEMP1* locus has previously been associated with height [76]. The authors found through Mendelian randomisation that the negative correlation between height and CTS likely reflects a causal relationship in which greater genetically-determined height diminishes the risk of CTS development [104], and hypothesised that altered skeletal growth affects the bony anatomy of the wrist, resulting in an anatomical configuration that is more likely to exert pressure on the median nerve. Moreover, fibulin-3 was also consistently highly expressed in the connective tissues surrounding the median nerve of CTS patients, spatially implicating the protein based on its prevalence in a disease tissue of interest [104]. Collectively, these findings provide evidence for a putative functional role of fibulin-3 in CTS pathogenesis, with at least some of the genetic risk of CTS likely arising from alteration of both the skeletal and connective tissue environment of the median nerve as it transits the carpal tunnel, which may in turn contribute to a greater predisposition to nerve entrapment [104].

#### 4.2.7. Childhood Ear Infection

Another *EFEMP1* GWAS association that may have a strong anatomical basis is susceptibility to childhood ear infections (CEIs) [105], a tendency that could in part relate to the specific shape of the eustachian tube [105]. It may be that the putative role of fibulin-3 in the development of cartilaginous structures is somehow implicated in this complex trait. Indeed, the authors did find that many of the variants associated with CEI were collectively involved in embryonic development [105], suggesting that there may be some cumulative effect of various development-related loci that manifests as an anatomically-based susceptibility to ear infection, of which *EFEMP1* genetic variation could be a contributing factor.

#### 4.2.8. Biliary Atresia

*EFEMP1* has also been associated with biliary atresia (BA), a condition that involves pathological fibrosing obliteration of the extrahepatic biliary tree. Aberrant upregulation of fibulin-3 in disease-associated tissues, including BA patient liver cholangiocytes and vascular smooth muscle cells, has been demonstrated [106]. Although it remains unclear if such upregulation represents a causal, consequential, or collateral phenomenon in cholestatic disease, given that fibulin-3 is normally more highly expressed in both cholangiocytes and portal fibroblasts relative to other liver cell types [106], the question of whether fibulin-3 is implicated in the pathogenesis of BA may warrant further investigation.

#### 4.2.9. Cancer

GWAS studies have also found *EFEMP1* variants to be associated with increased risk of two tumour types: glioma [107,108] and breast cancer [109]. Fibulin-3 expression changes have been detected in numerous cancers. It is upregulated in glioma [110], glioblastoma [93,111], osteosarcoma [112,113], and cervical cancer [114]; downregulated in carcinoma of the endometrium [115], prostate [116,117], lung [118,119,120,121,122], breast [123,124], colon [125,126,127,128], liver [129,130,131], nasopharynx [132], and thyroid [133]; and variably expressed in gastric [134,135] and ovarian [136,137,138] carcinomas. The role of fibulin-3 in the biology of these tumours has been extensively investigated, a full discussion of which is beyond the scope of this review, but the overarching picture emerging from the literature is that fibulin-3 has extensive, but highly context-dependent, effects in cancer, exerting both pro- and anti-tumour actions across many different tumour types. It is unlikely that fibulin-3 functions in isolation, but rather functions in concert with numerous members of the tumour extracellular environment, some of which may also modulate its activity, giving rise to tumour-type specific actions.

It is worth highlighting that some of the strongest evidence implicating fibulin-3 causally in tumour progression has arisen in the context of glioma, conceptually consistent with the implication of *EFEMP1* in a genetic susceptibility to glioma via GWAS. Fibulin-3 expression is absent from normal brain tissue [93] but is upregulated in human glioma [110] and glioblastoma [93,111], where it co-localises with tumour blood vessels [93]. This de novo expression is thought to be driven at the post-transcriptional level by downregulation of the inhibitory microRNA miR-338-5p, which normally suppresses fibulin-3 expression [111]. One further aspect of fibulin-3 pro-tumour signalling in glioma is thought to involve its interaction with TIMP-3. Fibulin-3 inhibition of TIMP-3 relieves TIMP-3 suppression of ADAM17, which subsequently may enhance glioma angiogenesis via downstream activation of endothelial tabulation [93] and pro-invasive NF-κB signalling [139]. Fibulin-3 also drives glioma invasiveness [110], a process mechanistically underpinned by modulation of the ECM surrounding the tumour through increasing expression of MMP2, MMP9, and ADAMTS-5 [92]. Such findings are illustrative of how the effects of fibulin-3 in cancer more generally occur via diverse actions on numerous cell types within the tumour microenvironment, adding an additional layer of complexity to its signalling functionality.

This clear tumour-driving effect has highlighted fibulin-3 as a putative therapeutic target in glioma/glioblastoma, in which it may be possible to impair tumour progression through suppression of fibulin-3 activity at either the gene, mRNA, or protein level (Figure 6).

Some of the anti-glioma therapeutic potential of metformin [140,141,142,143] appears to lie in its ability to downregulate fibulin-3 mRNA, causing suppression of MMP2 expression [144]. This action was associated with reduced invasiveness of human glioma cells [144], suggesting metformin may suppress tumour progression at least in part by inhibiting the ECM remodelling that underlies a metastasis-conducive tumour micro-environment through regulation of fibulin-3 expression.

Post-transcriptional regulation of fibulin-3 using microRNA has also been highlighted as a putative therapeutic strategy [111]. miR-338-5p-mediated downregulation of fibulin-3 mRNA decreases glioblastoma cell metastatic behaviour and increases their apoptosis [111]. This miRNA is downregulated in human glioblastoma tissues and cells in accordance with fibulin-3 upregulation, suggesting loss of its function may contribute to tumour progression and thus restoration of miR-338-5p could yield substantial therapeutic benefits.

The fibulin-3 protein could also be targeted directly. An anti-fibulin-3 monoclonal antibody (mAb) targeting a 23 amino acid signalling epitope of the protein has been developed that successfully inhibits fibulin-3-mediated activation of key glioblastoma intracellular signalling pathways (Notch, NF-κB, ADAM17). In a murine model, this treatment leads to increased tumour cell apoptosis; reduced tumour growth, invasion and vascularisation in xenograft glioblastomas; along with increased survival [145]. Clearly, extensive further investigation would be required to translate this into a clinically realistic approach, but this is undoubtedly an important proof-of-principle finding.

#### 4.2.10. *EFEMP1* rs3791679 Variant

From the collective analysis of these GWAS findings, *EFEMP1* genetic variants manifest functionally similar changes across numerous body systems that can contribute to the emergence of distinct phenotypes. Functional commonality in the effect of *EFEMP1* variants across such diverse pathologies is also supported by the fact that the same variant (rs379679) has been linked to many of these traits. Though one must be careful not to exaggerate the significance of a specific genetic variant being repeatedly detected in numerous studies, because this can represent a self-fulfilling artefact of the specific assays used to sequence gene variants, given that this specific variant resides in an enhancer region of the *EFEMP1* gene and has been predicted to have transcription factor-binding capability [83], it seems reasonable to infer that it has some degree of functional impact on fibulin-3 that could be worthy of further investigation. The variant has been associated with anthropometric traits (height [76], waist circumference (WC) [83], FVC [85]); glaucoma endophenotypes (IOP [89] and OCA [88]); abnormalities of connective tissue (joint hypermobility [82], CTS [104], and VVs [101]); and glioma [107]. Notably, some of these conditions have also been found to involve increased fibulin-3 expression levels (CTS [104] and glioma [107]), so it is possible that this specific SNP in the enhancer region leads to pathological *EFEMP1* overexpression. However, how this mechanistically leads to downstream disease remains to be elucidated.

## 5. Outlook: Fibulin-3 as a Regulator of ECM Enzymatic Equilibrium

Fibulin-3 is a molecule of increasing interest, given its highly diverse profile of pathophysiological associations. Unifying many of its multi-system effects appears to be its role as an important modulator of ECM biology. Fibulin-3 may contribute to the development, structural integrity, and emergent functionality of connective tissues across a range of anatomical locations. Abnormalities in its role as both a structural component and enzyme regulator in the ECM may also underpin its implication in other pathologies, including the RPE apoptosis and choroidal neovascularisation seen in macular degeneration, and the capacity for tumour cell invasion and metastasis in cancer. What remains less clear is how fibulin-3 functional variations arise from alterations at the genetic level (either mutations or SNPs). Such changes could relate to quantitative changes in fibulin-3 expression levels in specific cell types or to structural alterations that perturb either its own primary function or downstream functions secondary to changes in it its protein–protein interactions.

Over the last two decades, fibulin-3 has emerged as a molecule that plays a central role in connective tissue biology, human development, and disease. The advent of GWAS, in particular, has demonstrated its association with a plethora of human diseases in various organ systems. Notably, the fact that fibulin-3 is tractable to pharmaceutical targeting means that it is being increasingly investigated as a therapeutic target in the field of cancer biology. Given the ubiquity of fibulin-3 in human disease, these exciting developments will doubtless have repercussions for researchers in other medical fields.

## Figures and Tables

**Figure 1 biomolecules-10-01294-f001:**
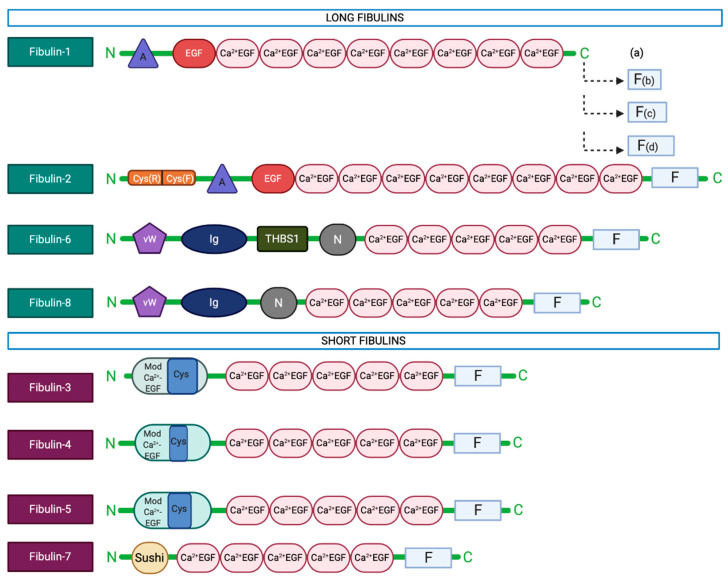
Fibulin family protein structures. A, anaphylatoxin-like domain; EGF, epidermal growth factor-like domain; Ca^2+^-EGF, calcium-binding EGF-like module; F, fibulin module; Cys(R) and Cys(F), cysteine-rich and cysteine-free sections of fibulin-2 C-terminal domain, respectively; vW, von Willebrand motif; Ig, immunoglobulin-like domain; THBS1, thrombospondin-1 motif; N, nidogen-like domain; Mod-Ca^2+^-EGF, modified Ca^2+^-EGF; Sushi, Sushi domain.

**Figure 2 biomolecules-10-01294-f002:**
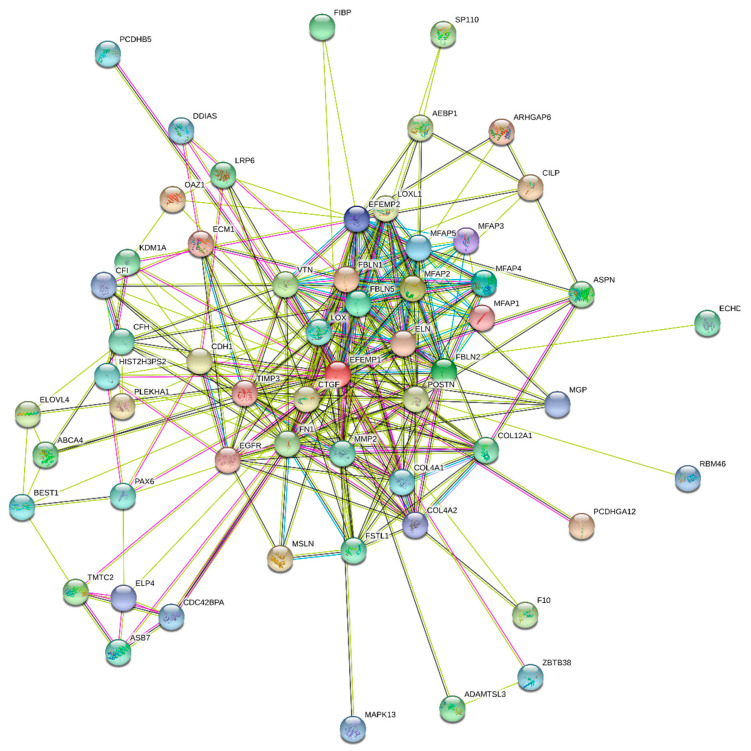
Search Tool for the Retrieval of Interacting Genes (STRING)-based analysis of human fibulin-3 (UniProt ID: Q12805) using the minimum required interaction score of 0.4 (medium confidence). STRING generates a network of predicted associations based on predicted and experimentally validated information on the interaction partners of a protein of interest. In the corresponding network, the nodes correspond to proteins, whereas the edges show predicted or known functional associations. Seven types of evidence are used to build the corresponding network, indicated by the differently coloured lines: a green line represents neighborhood evidence; a red line—the presence of fusion evidence; a purple line—experimental evidence; a blue line—co-occurrence evidence; a light blue line—database evidence; a yellow line—text mining evidence; and a black line—co-expression evidence.

**Figure 3 biomolecules-10-01294-f003:**
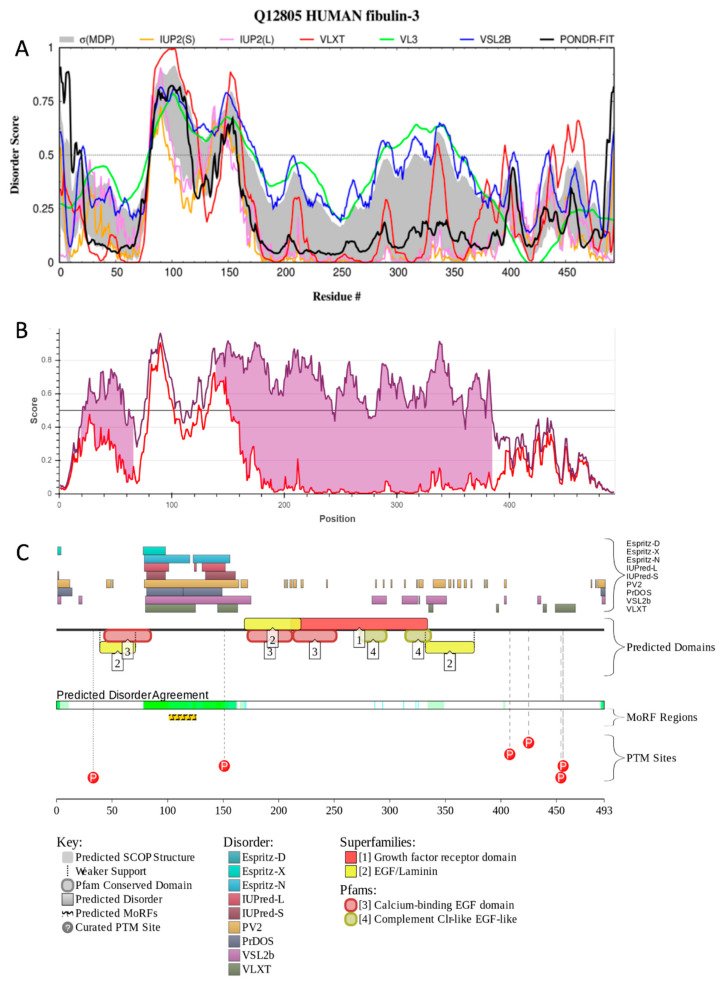
Intrinsic disorder status and disorder-based functional sites of human fibulin-3. (**A**) Multifactorial analysis of the intrinsic disorder predisposition of human fibulin-3 (UniProt ID: Q12805). Disorder profile is generated by a DiSpi web crawler designed to aggregate the results from a number of well-known disorder predictors: PONDR^®^ VLXT [38], PONDR^®^ VL3 [41], PONDR^®^ VLS2 [43], PONDR^®^ FIT [44], IUPred2 (Short), and IUPred2 (Long) [37,45]. This tool enables the rapid generation of disorder profile plots for individual polypeptides, as well as arrays of polypeptides. (**B**) Evaluation of the redox sensitivity of human fibulin-3 (i.e., the presence of cysteine-containing regions capable of the disorder-to-order or order-to-disorder transitions associated with changes in the redox state of the environment). (**C**) Functional disorder analysis of human fibulin-3 by the D^2^P^2^ platform (http://d2p2.pro/) [36]. At the top of this plot, there is a side-by-side comparison of seven disorder predictors (Espritz-D, Espritz-X, Espritz-N, IUPred-L, IUPred-S, PV2, PrDOS, PONDR^®^ VSL2b, and PONDR^®^ VLXT); bars indicate positive hits for disorder prediction. Below these coloured bars there are two bars showing positions of predicted SCOP (Structural Classification of Proteins) domains and conserved Pfam domains. The middle of each plot contains a bar labeled ‘Predicted Disorder Agreement’, presenting the level of agreement between all of the disorder predictors, which is shown as green colour intensity in an aligned gradient bar below the stack of predictions. Below the disorder agreement line, disorder-based binding regions (i.e., MoRFs) predicted by ANCHOR are displayed as yellow blocks with zigzag infill. Finally, the bottom of the plot represents positions of phosphorylation sites, which are shown by red circles containing the letter P (phosphorylation).

**Figure 4 biomolecules-10-01294-f004:**
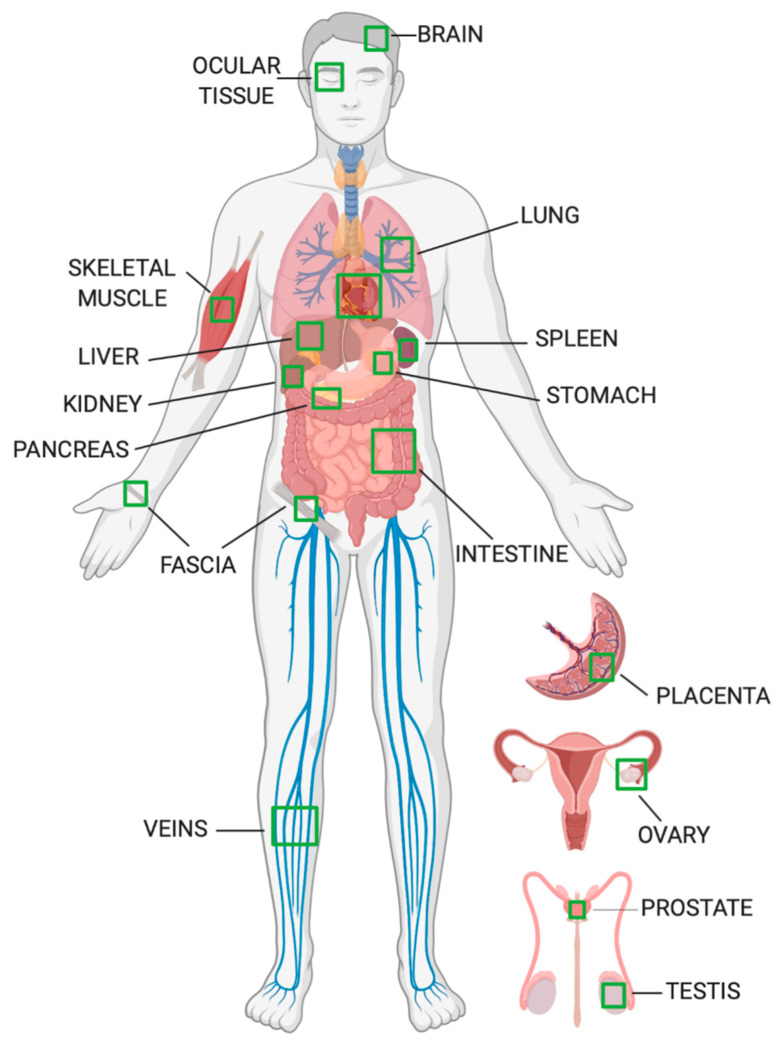
Anatomical distribution of fibulin-3 expression in humans.

**Figure 5 biomolecules-10-01294-f005:**
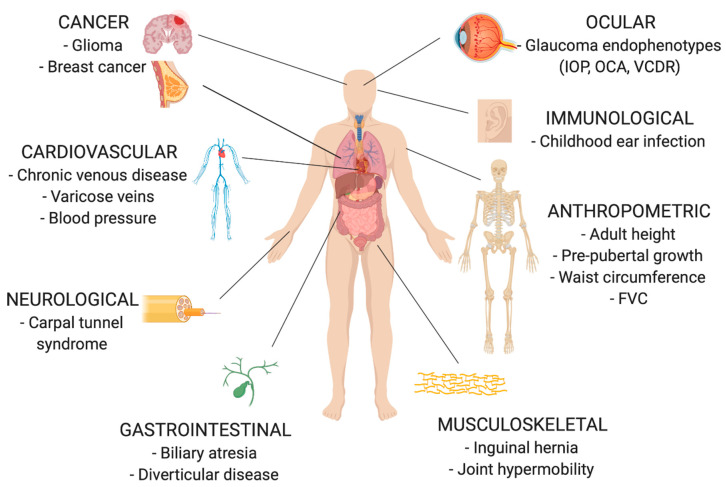
Pathophysiological associations of *EFEMP1* discovered in genome-wide association studies.

**Figure 6 biomolecules-10-01294-f006:**
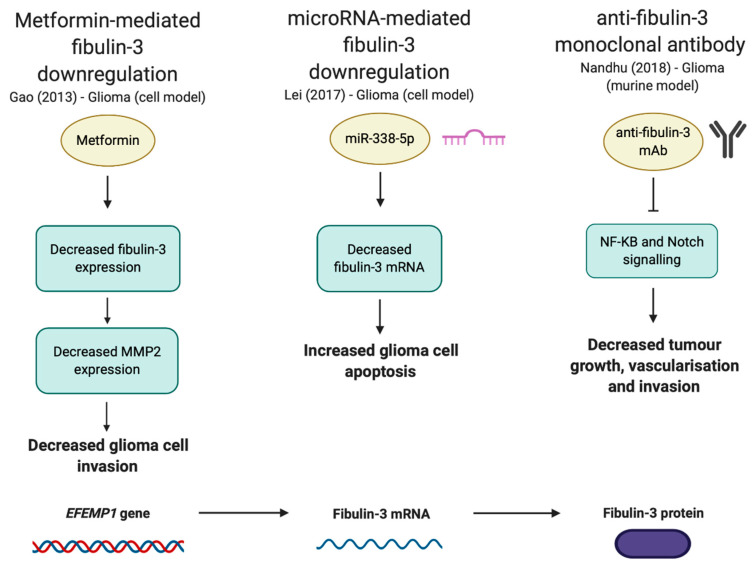
Putative fibulin-3-oriented therapeutic strategies for glioma.

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
