# Peer review of "The Pathophysiological Significance of Fibulin-3"

_biomolecules, 2020, doi:10.3390/biom10091294_

Round 1

Reviewer 1 Report

The review entitled: "The Pathophysiological Significance of Fibulin-3" is a well written, very extensive review on its topic. All pathophysiological implications with fibulin-3 are included in a comprehensive and upto date way. Nearly all relevant papers of a wide range of authors are cited. The figures are elegant and informative. There is nothing, thats need to be corrected.

Author Response

We are grateful to the Reviewer for reviewing our manuscript and for the positive comments. 

Reviewer 2 Report

This review provide insightful and discussion of fibulin-3 in different diseases. References are up-to-date. This reviewer would suggest the publication of this manuscript in the current form. 

Author Response

(The authors gave the same response as above.)

Reviewer 3 Report

The authors have reviewed the function of fibulin 3 in biology as well as its role in human diseases. In general, the review article is a comprehensive presentation of fibulin-3, but It can be further improved. I have following suggestions to improve the manuscript:

  1. Way too much space and references are devoted to intrinsically disordered proteins (IDPs) topic. If one looks at the IDPs-topic, it has very little/almost no relevance to fibulin-3 gene.

  1. The same criticism applies to genome-wide association studies (GWASs). The GWASs are notoriously studies in the medical literature that cannot be reproduced and have no/very little clinical relevance. I would devote far less attention to GWAs-studies linking fibulin-3 to certain disease unless there is clear biochemical evidence that fibulin-3 plays a definite role in the certain disease.

  1. The figures are representative. However, I would like to see one where the expression pattern of fibulin-3 is described.

  1. The general function of fibulins is form protective layer around elastin in elastic blood vessels. I would devote far more attention to this interaction and the fact that fibulin-3 does not seem to play a role in it and then to explain why its function/expression pattern is different from other, “classical” fibulins.
